# UpFusion: Novel View Diffusion from Unposed Sparse View Observations

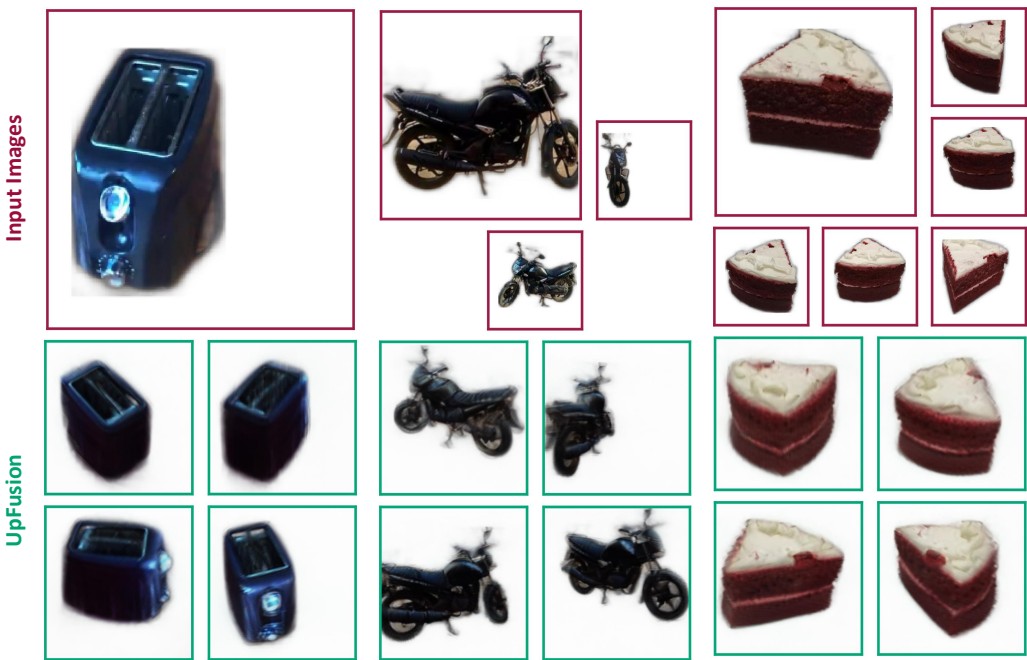

Figure 1: **3D Inference from Unposed Sparse views**. Given a sparse set of input images without associated camera poses, our proposed system *UpFusion* allows recovering a 3D representation and synthesizing novel views. *Top:* 1, 3, or 6 input images of an object. *Bottom:* Synthesized novel views using our approach.

## Abstract

We propose UpFusion, a system that can perform novel view synthesis and infer 3D representations for an object given a sparse set of reference images *without* corresponding pose information. Current sparse-view 3D inference methods typically rely on camera poses to geometrically aggregate information from input views, but are not robust in-the-wild when such information is unavailable/inaccurate. In contrast, UpFusion sidesteps this requirement by learning to implicitly leverage the available images as context in a conditional generative model for synthesizing novel views. We incorporate two complementary forms of conditioning into diffusion models for leveraging the input views: a) via inferring query-view aligned features using a scene-level transformer, b) via intermediate attentional layers that can directly observe the input image tokens. We show that this mechanism allows generating high-fidelity novel views while improving the synthesis quality given additional (unposed) images. We evaluate our approach on the Co3Dv2 dataset and demonstrate the benefits of our method over pose-reliant alternates. Finally, we also show that our learned model can generalize beyond the training categories, and hope that this provides a stepping stone to reconstructing generic objects from in-the-wild image collections.

# 1 INTRODUCTION

The long-standing problem of recovering 3D objects from 2D images has witnessed remarkable recent progress. In particular, recent neural field-based methods (Mildenhall et al., 2020) excel at recovering highly detailed 3D models of objects or scenes given densely sampled multi-view observations. However, in real-world scenarios such as casual capture settings and online marketplaces, obtaining dense multi-view images is often impractical. Instead, only a limited set of observed views may be available, often leaving some aspects of the object unobserved. With the goal of reconstructing similarly high-fidelity 3D objects in these settings, several learning-based methods (Yao et al., 2018; Yu & Gao, 2020; Zou et al., 2023) have pursued the task of sparse-view 3D inference. While these methods can yield impressive results, they crucially rely on known accurate camera poses for the input images, which are often only available in synthetic settings or using privileged information in additional views, and are thus not currently applicable for in-the-wild sparse-view reconstruction where camera poses are not available.

In this work, we seek to overcome the limitation of requiring known camera poses and address the task of 3D inference given *unposed* sparse views. Unlike pose-aware sparse-view 3D inference methods which use geometry-based techniques to leverage the available input, we introduce an approach that can implicitly use the available views for novel-view generation. Specifically, we designate one of the input images as an anchor to define a coordinate frame, and adopt a scene-level transformer (Sajjadi et al., 2022) that implicitly incorporates all available input images as context to compute per-ray features for a desired query viewpoint. Utilizing these query-aligned features, we can train a conditional denoising diffusion model to generate novel-view images.

However, we observe that relying solely on query-aligned features learned from unposed input views does not fully utilize the available context. To further enhance the instance-specificity in the generations, we propose to also add 'shortcuts' via attention mechanism in the diffusion process to allow direct attending to the input view features during the generation. Furthermore, to enable generalization to unseen categories during training, we adopt a pretrained 2D foundation diffusion model (Rombach et al., 2022; Zhang & Agrawala, 2023) as initialization and adapt it to leverage the two forms of context-based conditioning. Finally, the novel view images synthesized from the learned diffusion model, despite high fidelity, may not guarantee 3D consistency. Therefore, we additionally extract 3D-consistent models via score-based distillation (Poole et al., 2022; Zhou & Tulsiani, 2023).

We present results using the challenging real-world dataset, Co3Dv2 (Reizenstein et al., 2021), which comprises multi-view sequences from 51 categories with 6-DoF pose variations. Given our unposed inference setup, we also introduce 'alignment invariant' versions of common evaluation metrics to account for the possible coordinate mismatch between the predicted and ground-truth 3D representations. We find that our approach allows extracting signal from the available unposed views, and that the performance improves with additional images, and that our system significantly improves over recent pose-aware methods relying on predicted camera poses. Finally, we also demonstrate the ability of our method to generalize beyond the training categories by showcasing its performance on unseen object classes.

# 2 RELATED WORK

**3D from Dense Multi-view Captures.** Multi-view observations of a scene naturally provide geometric cues for understanding its 3D structure, and this principle has been leveraged across decades to infer 3D from dense multi-view. Classical Multi-View Stereo (MVS) methods (Furukawa et al., 2015) leverage techniques such as structure from motion (SfM) (Schönberger & Frahm, 2016) to estimate camera poses for dense matching to 3D points. Recent neural incarnations (Mildenhall et al., 2020; Wang et al., 2021a) of these methods have further enabled breakthroughs in terms of the quality of the obtained dense 3D reconstruction. While these methods rely on classical techniques for camera estimation, subsequent approaches (Lin et al., 2021; Bian et al., 2023; Tian et al., 2023) have relaxed this requirement and can jointly estimate geometry and recover cameras. However, these methods are unable to predict unseen regions and crucially rely on densely-sampled images as input – a requirement our work seeks to overcome.

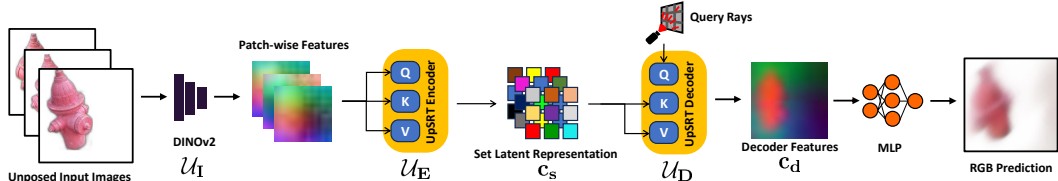

Figure 2: **UpSRT** performs novel view synthesis from a set of unposed images. UpSRT consists of an encoder, a decoder, and an MLP. The encoder takes encoded image features as inputs and outputs a set-latent representation $c_s$. The decoder takes query rays as inputs and attends to the set-latent representation to get features $c_d$, which are then fed into MLP to obtain final novel view RGB images. We make use of both $c_s$ and $c_d$ to provide conditional context to our model.

**Single-view to 3D.** On the other extreme from dense multi-view methods are approaches that aim to reconstruct a 3D representation from just a single view. While easily usable, developing such systems is highly challenging as it requires strong priors to recover unknown information. A common paradigm used to address this program is training models conditioned on encoded image features to directly predict 3D geometry (*e.g.,* voxels (Girdhar et al., 2016), meshes (Wang et al., 2018; Gkioxari et al., 2019; Ye et al., 2021), point clouds (Fan et al., 2017), or implicit functions (Mescheder et al., 2019; Xu et al., 2019; Cheng et al., 2023)). However, given the uncertain nature of the task, these methods have regression-based objectives which limits their generation quality. More recently, there has been growing interest in distilling large text-to-image diffusion models (Song et al., 2020; Saharia et al., 2022; Rombach et al., 2022) to generate 3D representations (Poole et al., 2022; Wang et al., 2023a;b; Chen et al., 2023). Building upon their advances, several distillation-based (Liu et al., 2023b; Qian et al., 2023; Deng et al., 2023; Melas-Kyriazi et al., 2023; Tang et al., 2023; Xu et al., 2022) and distillation-free (Liu et al., 2023a;c) single image to 3D methods were proposed. While these methods can infer detailed 3D, they cannot benefit from additional information provided by extra posed or unposed views. Moreover, as they hallucinate details in unobserved regions, the reconstructed object may significantly differ from the one being imaged. If a user aims to faithfully capture a specific object of interest in detail, single-view methods are fundamentally ill-suited for this task.

**Sparse-view to 3D.** With the goal of reducing the burden in the multi-view capture process while still enabling detailed capture of specific objects of interest, there has been a growing interest in spare-view 3D inference methods. By leveraging the benefits of both multi-view geometry and learning, regression-based methods achieve 3D consistency by using re-projected features obtained from input views (Reizenstein et al., 2021; Wang et al., 2021b; Yu et al., 2021). However, the results tend to be blurry due to the mean-seeking nature of regression methods under uncertainty. To improve the quality of generations, another stream of work (Chan et al., 2023; Rombach et al., 2021; Kulhánek et al., 2022; Zhou & Tulsiani, 2023) formulate the problem as a probabilistic generation task. These methods achieve better perceptual quality, yet usually require precise pose information, which is often not practically available. To overcome this issue, one may either consider leveraging recent sparse-view pose estimation methods (Sinha et al., 2023; Zhang et al., 2022) in conjunction with state-of-the-art novel-view synthesis methods, or consider methods that optimizes poses jointly with the objective of novel-view synthesis (Smith et al., 2023; Jiang et al., 2022). However, the computation of explicit poses may not always be robust, and we emprically show that this leads to poor performance. Closer to our approach, SRT (Sajjadi et al., 2022) and RUST (Sajjadi et al., 2023) allow novel view synthesis without explicit pose estimation (i.e., directly from unposed sparse views). However, their regression-based pipelines limit the quality of the synthesized outputs.

## 3 APPROACH

Our goal is to infer a 3D representation of an object given a sparse set of images. While prior works (Yu et al., 2021; Zhou & Tulsiani, 2023; Chan et al., 2023) typically aggregate information from the input views by using geometric projection and unprojection, these crucially rely on the availability

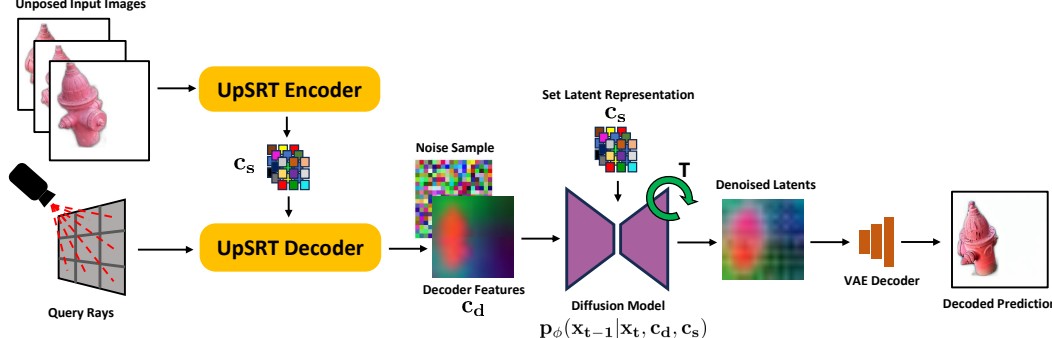

Figure 3: **UpFusion 2D** is the proposed conditional diffusion model performing novel view synthesis conditional on information extracted from a set of unposed images. To reason about the query view, Upfusion takes as additional inputs the view-aligned decoder features $c_d$ obtained from UpSRT decoder. To further allow the model to attend to details from input views, UpFusion condition on the set-latent representation $c_s$ via attentional layers.

of accurate camera poses which are not readily available in-the-wild. We instead aim to tackle the task of 3D inference given *unposed* sparse views.

Towards building a system capable of 3D inference in this unposed setting, we propose a mechanism for implicitly leveraging the available images as context when generating novel views. Specifically, we adapt Unposed Scene Representation Transformer (UpSRT) (Sajjadi et al., 2022), a prior work that leverages transformers as a mechanism for implicitly aggregating information from input views, and computes query-view-aligned features for view synthesis. However, instead of their mean-seeking regression objective which results in blurry renderings, we enable probabilistic sparse view synthesis by using the internal representations of UpSRT to condition a diffusion model to perform novel view synthesis. While our diffusion model can yield high-fidelity generations, the outputs are not 3D consistent. To obtain a consistent 3D representation, we then train instance-specific neural representations (Müller et al., 2022; Tang, 2022) which maximizes the likelihood of the renderings under the learned generative model. We detail our approach below, but first briefly review UpSRT and denoising diffusion models (Ho et al., 2020) that our work builds on.

### 3.1 PRELIMINARIES

#### 3.1.1 UNPOSED SCENE REPRESENTATION TRANSFORMER

Given a set of $N$ images $\mathcal{I} = \{I_1, I_2, ..., I_N\}$, UpSRT (Sajjadi et al., 2022) seeks to generate novel view images by predicting RGB color $r$ for any query ray $q$. As illustrated in figure 2, it first extracts patch-wise features for each image $I_i$ with an image encoder $\mathcal{U}_I$. Then, it uses an encoder transformer $\mathcal{U}_E$ to obtain a set latent representation $c_s$. Finally, it uses a decoder transformer $\mathcal{U}_D$ which attends to $v_c$, followed by an MLP, to predict the RGB color. In summary, the UpSRT workflow can be represented by the following equations:

$$c_s = \mathcal{U}_E(\{\mathcal{U}_I(\mathcal{I})\}), C(r) = \text{MLP}(\mathcal{U}_D(r|c_s)) \tag{1}$$

We pre-train an UpSRT model using a pixel-level regression loss and leverage it for subsequent generative modeling. While we follow a similar design, we make several low-level modifications from the originally proposed UpSRT architecture (*e.g.,* improved backbone, differences in positional encoding, *etc.*), and we expand on these in the appendix.

#### 3.1.2 DENOSING DIFFUSION

Denoising diffusion models (Ho et al. (2020)) seek to learn a generative model over data samples $x$ by learning to reverse a forward process where noise is gradually added to original samples. The

learning objective can be reduced to a denoising error, where a diffusion model $\epsilon_\phi$ is trained to estimate the noise added to a current sample $\boldsymbol{x}_t$:

$$\mathcal{L}_{DM} = \mathbb{E}_{\boldsymbol{x}_0, t, \epsilon \sim \mathcal{N}(0,1)}[\|\epsilon_t - \epsilon_\phi(\boldsymbol{x}_t, t)\|_2^2] \tag{2}$$

While the above objective summarizes an unconditional diffusion model, it can be easily adapted to learn conditional generative models $p(\boldsymbol{x}|\boldsymbol{y})$ by adding a condition $\boldsymbol{y}$ (such as a set of unposed images) to the input of the denoising model $\epsilon_\phi(\boldsymbol{x}_t, t, \boldsymbol{y})$.

## 3.2 Probabilistic View Synthesis using Sparse Unposed Views

We aim to learn a generative model over novel views of an object given a sparse set of unposed images. We note that there is an inherent ambiguity in defining the coordinate frame in which this query view is specified, and (partially) resolve this by using the first input image as an anchor to define the coordinate system. Given this, our goal is to learn the distribution $p(\mathbf{I}|\boldsymbol{\mathcal{I}}, \pi)$, where $\pi$ denotes a query pose, $\boldsymbol{\mathcal{I}}$ denotes the set of unposed images and $\mathbf{I}$ denotes the query-view image. Instead of learning the distribution directly in pixel space, we follow a common practice of instead learning this distribution in latent space $p(\boldsymbol{x}|\boldsymbol{\mathcal{I}}, \pi)$, using pre-trained encoders and decoders corresponding to this latent space (Rombach et al., 2022): $\boldsymbol{x} = \mathcal{E}(\mathbf{I}); \ \mathbf{I} = \mathcal{D}(\boldsymbol{x})$.

We model this probability distribution by training a conditional diffusion model which leverages the available unposed images as context, and seek to propose an architecture that embraces several desirable design principles. First, we note that such a diffusion model must be able to (implicitly) reason about the query view it is tasked with generating in the context of the available input, and leverage the UpSRT encoder-decoder framework to enable this. While the decoder features from UpSRT can ground the query-view generation, we note that these may abstract away the salient details in the input, and we propose to complement these by allowing the generative model to directly leverage the patch-wise latent features and more easily 'copy' content from input views. Lastly, to enable efficient training and generalization beyond training data, we propose to adapt off-the-shelf diffusion models for view-conditioned generation.

**View-aligned Features for Image Generation.** Given a target view $\pi$, we construct a set of rays $\mathcal{R}$ corresponding to a grid of 2D pixel locations in this view. We query the UpSRT decoder with this set of rays to obtain view-aligned decoder features $\boldsymbol{c}_d$ of the same resolution as the image latent $\boldsymbol{x}$. As illustrated in figure 3, these query-aligned features are concatenated with the (noisy) image latents to serve as inputs to the denoising diffusion model.

**Incorporating Direct Attention to Input Patches.** To allow the generation model to directly incorporate details visible in the input views, we also leverage the set-latent feature $\boldsymbol{c}_s$ representation extracted by the UpSRT encoder. Importantly, this representation comprises of per-patch features aligned with the input images and allows efficiently 'borrowing' details visible in these images. Unlike the view-aligned decoder feature which can be spatially concatenated with the noisy diffusion input, we condition on these set-latent features via attentional layers in the generation model.

**Adapting Large-scale Diffusion Models for Novel-view Synthesis.** Instead of training our generative model from scratch, we aim to take advantage of the strong priors learned by large diffusion models such as Stable Diffusion (Rombach et al. (2022)). To this end, we use a modified version of the ControlNet architecture (Zhang & Agrawala, 2023) to adapt a pre-trained Stable Diffusion model to incorporate additional conditionings $\boldsymbol{c}_d, \boldsymbol{c}_s$ for view generation.

**Putting it Together.** In summary, we reduce the task of modeling $p(\boldsymbol{x}|\boldsymbol{\mathcal{I}}, \pi)$ to learning a denoising diffusion model $p_\phi(\boldsymbol{x}|\boldsymbol{c}_d, \boldsymbol{c}_s)$, and leverage the ControlNet architecture to incorporate the two conditioning features and learn a denoising model $\epsilon_\phi(\boldsymbol{x}_t, t, \boldsymbol{c}_d, \boldsymbol{c}_s)$. More specifically, ControlNet naturally allows adding the spatial feature $\boldsymbol{c}_d$ as via residual connections to the spatial layers of the UNet in a pre-trained Stable Diffusion model. To incorporate the set-level features $\boldsymbol{c}_s$, we modify the ControlNet encoder blocks to use $\boldsymbol{c}_s$ in place of a text encoding (see appendix for details). We can train such a model using any multi-view dataset, where we train the denoising diffusion model to generate the underlying image from a query view given a variable number of observed input views.

### 3.3 Inferring 3D Consistent Representations

While the proposed conditional diffusion model can provide high-fidelity renderings from query views, the generated views are not 3D consistent. To obtain a 3D representation given the inferred distribution over novel views, we subsequently optimize an instance-specific neural representation. Towards this, we follow SparseFusion (Zhou & Tulsiani, 2023) which seeks neural 3D modes by optimizing the likelihood of their renderings by adapting a Score Distillation Sampling (SDS) (Poole et al., 2022) loss to view-conditioned generative models.

Specifically, we optimize a neural 3D representation $g_\theta$ by ensuring its renderings have high likelihood under our learned distribution $p(\mathbf{I}|\mathcal{I}, \pi)$. We do so by minimizing the difference between the renderings of the instance-specific neural model and the denoised predictions from the learned diffusion model. Denoting by $g_\theta(\pi)$ the rendering of the neural 3D representation from viewpoint $\pi$, and by $\hat{\boldsymbol{x}}_0$ the denoised prediction inferred from the learned diffusion model $\epsilon_\phi(\boldsymbol{x}_t; t, \boldsymbol{c}_d, \boldsymbol{c}_s)$, the training objective can be specified as:

$$\mathcal{L}_{3D} = \mathbb{E}_{t,\epsilon,\pi}[\|g_\theta(\pi) - \mathcal{D}(\hat{\boldsymbol{x}}_0)\|^2] \tag{3}$$

Unlike SparseFusion (Zhou & Tulsiani, 2023) which additionally uses a rendering loss for the available input views using known cameras, we rely only on the above denoising objective for optimizing the underlying 3D representation given unposed input views.

### 3.4 Training Details

We follow a multi-stage training procedure to optimize our models. We first train the UpSRT model separately using a reconstruction loss on the color predicted for query rays given the set of reference images $\mathbf{I}$. Then, we train the denoising diffusion model while using the conditioning information from the pre-trained UpSRT, which is frozen in this stage.

To enable the usage of classifier-free guidance (Ho & Salimans, 2021) during inference, we train our diffusion model in the unconditional mode for a small fraction of the time. We do this by following the condition dropout procedure used in (Brooks et al., 2023; Liu et al., 2023b) that randomly replaces the conditioning information with null tokens (for more details, see B.2).

Once the diffusion model is trained, we can extract a 3D representation for an object by optimizing an Instant-NGP (Müller et al., 2022; Tang, 2022) using the neural mode seeking objective discussed in section 3.3. We use DDIM (Song et al., 2020) for fast multi-step denoising. Inspired by Wang et al. (2023b), we follow an annealed time schedule for score distillation. We also use some regularization losses while training the NeRF as used in Zhou & Tulsiani (2023). For more details, please refer to section B.3.

## 4 Experiments

### 4.1 Experimental Setup

#### 4.1.1 Dataset

We train and evaluate our models on Co3Dv2 (Reizenstein et al., 2021), a large-scale dataset with real multi-view images of objects from 51 categories. Following (Zhang et al., 2022; Lin et al., 2023), we train our model on 41 categories and hold out 10 categories to test the ability of our method to generalize to unseen categories. We use the *fewview-train* split for training and *fewview-dev* split for evaluation. We limit our focus to modelling only objects and not their backgrounds. To this end, we create a white background for our objects by using the masks available in the dataset. As our full method (as well as some baselines) optimize instance-specific neural representations, which can take 1hr per instance, we limit our evaluations to 5 object instances per category.

We note that popular state-of-the-art single-view baselines are trained on Objaverse (Deitke et al., 2023). Hence, to allow fair comparison, we fine-tune a version of our model (which are already pre-trained on Co3Dv2) on Objaverse renderings as well. We denote versions of our model fine-tuned on Objaverse with † as a superscript (for example, UpFusion† (3D)).

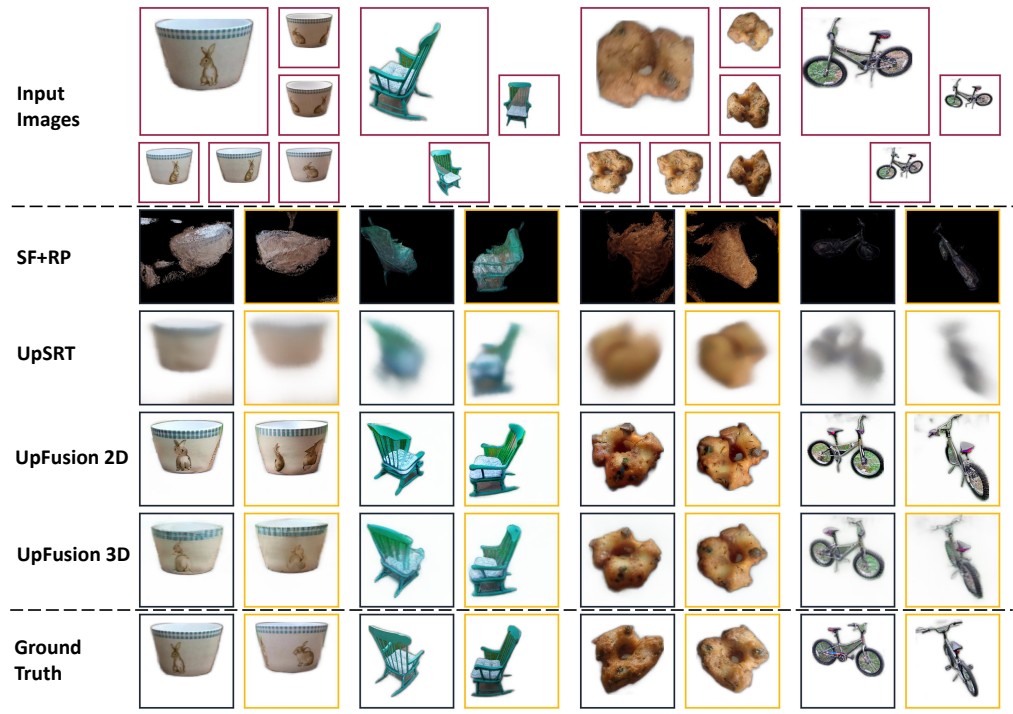

Figure 4: **Qualitative comparison with sparse-view baselines** We compare UpFusion with baseline methods using 3 and 6 unposed images as inputs. SparseFusion fails to capture the correct geometry, due to the imperfect camera poses estimated by RelPose++. UpSRT generates blurry results due to the nature of regression-based methods. On the contrary, UpFusion 2D synthesizes sharp outputs with correct object poses. UpFusion 3D further improves the 3D consistency.

| Type | Method | PSNR-A (↑) | | | SSIM-A (↑) | | | LPIPS-A (↓) | | |
|------|--------|------|------|------|------|------|------|------|------|------|
| | | 1V | 3V | 6V | 1V | 3V | 6V | 1V | 3V | 6V |
| Posed | SparseFusion (GT) | — | 22.41 | 24.02 | — | 0.79 | 0.81 | — | 0.20 | 0.18 |
| Unposed | SparseFusion (RelPose++) | — | 17.76 | 17.12 | — | 0.67 | 0.64 | — | 0.30 | 0.33 |
| | UpSRT | 16.84 | 17.75 | 18.36 | 0.73 | 0.74 | 0.75 | 0.34 | 0.32 | 0.31 |
| | UpFusion (2D) | 16.54 | 17.12 | 17.41 | 0.71 | 0.72 | 0.73 | 0.23 | 0.22 | 0.22 |
| | UpFusion (3D) | **18.17** | **18.68** | **18.96** | **0.75** | **0.76** | **0.76** | **0.22** | **0.21** | **0.21** |

Table 1: **Sparse-view synthesis evaluation on seen categories (41 categories).** We conduct comparisons using 5 samples per category and then and report the average across these. UpFusion performs favorably against baseline methods, and demonstrates the capability to improve the results when more views are provided. Moreover, UpFusion 3D consistently improves the results from UpFusion 2D.

### 4.1.2 EVALUATING VIEW SYNTHESIS IN UNPOSED SETTINGS

We are interested in evaluating our performance using standard view-synthesis metrics such as PSNR, SSIM, and LPIPS (Zhang et al. (2018)). However, these pixel-aligned metrics are not well suited for evaluating unposed view synthesis due to the fundamental ambiguities between the coordinate systems of the ground-truth and prediction. In particular, given unposed images, there can be an ambiguity up to a similarity transform between the coordinate frames of the reconstruction and prediction. While anchoring the coordinate orientation to the first camera reduces this uncertainty, we still need to consider scaling and shift between predictions and ground truth.

We highlight this issue in Figure 12, where we observe that despite generally matching the ground truth, the prediction is misaligned in pixel space. To circumvent this issue, we compute *aligned*

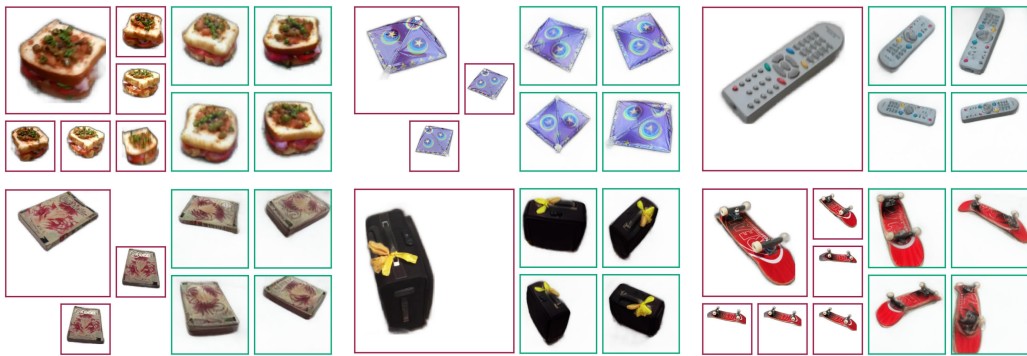

Figure 5: **Generalization beyond training categories.** We show results for UpFusion (3D) across object categories *not* seen in training. For each instance, we present the 1, 3, or 6 unposed input views (left), as well as 4 novel view renderings (right). We observe that despite not being trained on these categories, UpFusion is able to accurately infer the underlying 3D structure and generate detailed novel views.

versions of the standard image reconstructions metrics (PSNR-A, SSIM-A, and LPIPS-A) by first optimizing for an affine image warping transform $\mathcal{W}_A$ that best matches a predicted image to its corresponding ground truth and then computing the metric. In other words, we evaluate aligned metrics as $\min_{W_A} \mathcal{M}(\mathcal{W}_A(x), y)$, where $\mathcal{M}$ is a metric, $x$ is a predicted image and $y$ is the ground truth image. In practice, for expediency, we compute the optimal transform for minimizing a pixel-wise L2 error instead of computing a per-metric warp.

### 4.1.3 BASELINES

We highlight the benefits of our approach by comparing it to prior pose-dependent and unposed novel-view generation techniques. Specifically, we compare our 2D diffusion model ('UpFusion 2D') and obtained 3D representations ('UpFusion 3D') against the following baselines:

*SparseFusion* (Zhou & Tulsiani, 2023) is a current state-of-the-art method for pose-dependent sparse-view inference on Co3Dv2. We compare against its performance when using a recent sparse-view pose estimation system RelPose++ (Lin et al., 2023), and also report its performance using GT camera poses as an upper bound.

*UpSRT.* As a representative approach for view synthesis from unposed images, we compare against the prediction from the UpSRT (Sajjadi et al., 2022) backbone used in our approach.

*FORGE* (Jiang et al., 2022) is a method that jointly optimizes for poses while being trained on a novel-view synthesis objective. As FORGE uses the GSO dataset (Downs et al., 2022) to demonstrate its generalization capability, we compare it against our Objaverse fine-tuned UpFusion[†] (3D).

*Single-view methods.* To highlight the benefit of using more input views, we compare UpFusion[†] (3D) to two representative state-of-the-art single-view baselines: Zero-1-to-3 (Liu et al., 2023b) and One-2-3-45 (Liu et al., 2023a). For Zero-1-to-3, we include comparisons with two versions – the original version which uses SJC (Wang et al., 2023a) and the highly optimized threestudio implementation (Guo et al., 2023) (which uses additional tricks to aid 3D distillation). We compare against these baselines on the GSO dataset.

### 4.2 RESULTS

#### 4.2.1 NOVEL-VIEW SYNTHESIS ON CO3DV2

**Comparisons against Sparse-view Methods.** We compare UpFusion with baseline methods on the categories seen during training, as shown in Table 1. UpFusion performs favorably against both UpSRT and unposed SparseFusion. Furthermore, UpFusion consistently improves the prediction when more views are provided. However, there is still room for improvement compared to the methods using ground-truth poses. In figure 4, we qualitatively present the novel view synthesis

| Method | PSNR-A ($\uparrow$) | | | SSIM-A ($\uparrow$) | | | LPIPS-A ($\downarrow$) | | |
|---|---|---|---|---|---|---|---|---|---|
| | 1V | 3V | 6V | 1V | 3V | 6V | 1V | 3V | 6V |
| UpSRT | 16.75 | 17.57 | 18.06 | 0.73 | 0.74 | 0.74 | 0.35 | 0.33 | 0.32 |
| UpFusion (2D) | 16.33 | 17.04 | 17.38 | 0.70 | 0.71 | 0.72 | 0.25 | 0.23 | 0.23 |
| UpFusion (3D) | **18.27** | **18.83** | **19.11** | **0.75** | **0.76** | **0.76** | **0.23** | **0.22** | **0.22** |

Table 2: **Sparse-view synthesis evaluation on unseen categories (10 categories).** We conduct comparisons using 5 samples per category and report the average across these. We observe a comparable performance to the results on seen categories.

| # Input Views | Method | PSNR ($\uparrow$) | SSIM ($\uparrow$) | LPIPS ($\downarrow$) |
|---|---|---|---|---|
| 1V | Zero-1-to-3 (SJC) | 18.72 | 0.90 | 0.12 |
| | Zero-1-to-3 (TS) | 21.71 | **0.91** | 0.09 |
| | One-2-3-45 | 17.77 | 0.87 | 0.15 |
| | UpFusion† (3D) | 20.52 | 0.89 | 0.12 |
| 6V | FORGE | 17.40 | 0.88 | 0.15 |
| | UpFusion† (3D) | **22.51** | **0.91** | **0.08** |

Table 3: **Novel-view synthesis evaluation on GSO.** We compare UpFusion 3D to single-view baselines as well as a sparse-view pose-optimization baseline on GSO dataset which is out of distribution for all methods.

results. SparseFusion can capture some details visible in the input views but largely sufferrs due to the error in input poses. UpSRT, on the other hand, can robustly generate coarse renderings, but is unable to synthesize high-fidelity outputs from any viewpoints. Our 2D diffusion model, UpFusion 2D, generates higher fidelity images that improve over the baselines in the perceptual metrics. Finally, the 3D-consistent inferred representation Upfusion3D yields the best results.

**Characterizing Generalization.** As UpFusion is trained upon a pre-trained large-scale diffusion model providing strong general priors, the learned novel view synthesis capability is expected to be generalized to categories beyond training. We evaluate UpFusion on 10 unseen categories, as shown in Table 2. Encouragingly, we find that the performance does not degrade compared to the results on seen categories and believe this highlights the potential of our approach to perform in-the-wild sparse-view 3D inference. We also depict some qualitative results on unseen objects in Figure 5.

### 4.2.2 Novel-view synthesis on GSO

We compare UpFusion† (3D) to two state-of-the-art single-view baselines (Zero-1-to-3 and One-2-3-45) and a sparse-view baseline (FORGE) on 20 randomly sampled instances from the GSO dataset. For Zero-1-to-3, we compare with both the original SJC implementation and threestudio (TS) implementation. From table 3, we can observe that UpFusion† (3D) while using 6 inputs views is able to outperform all baselines. This demonstrates the ability of our method to effectively incorporate more information when additional views are available, which single-view baselines cannot. Moreover, we can see that our model significantly outperforms FORGE, which also uses 6 input views, and we believe this is because our approach allows bypassing explicit pose prediction which can lead to inaccurate predictions. Qualitative comparisons in Figure 6 further demonstrates the effectiveness of our approach in utilizing information from multiple unposed images.

## 5 Discussion

We presented UpFusion, an approach for novel-view synthesis and 3D inference given unposed sparse views. While our approach provided a mechanism for effectively leveraging unposed images as context, we believe that several challenges still remain towards the goal of sparse-view 3D inference in-the-wild. In particular, although our approach allowed high-fidelity 2D generations, these are not always precisely consistent with the details in the (implicitly used) input views. Moreover, while our approach's performance does improve given additional context views, it does not exhibit

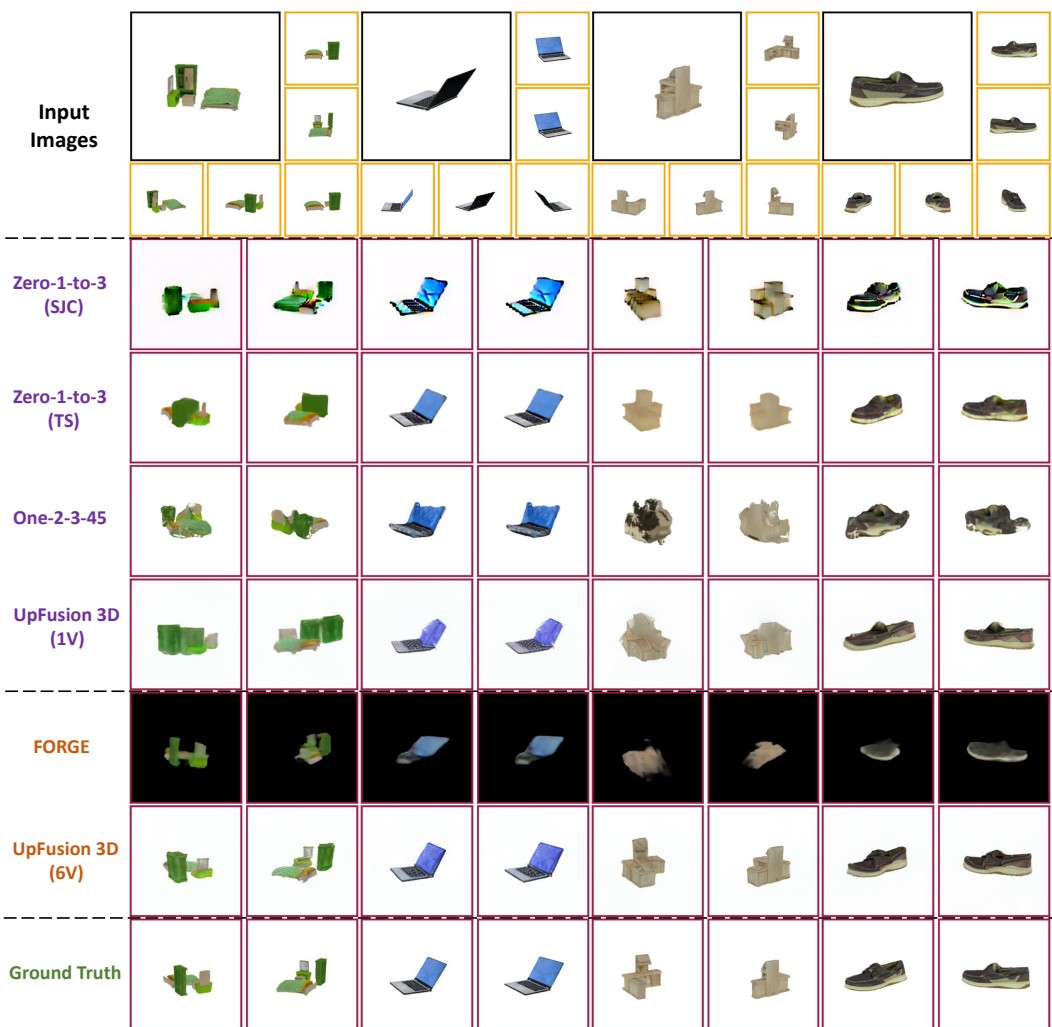

Figure 6: **Qualitative comparison on GSO.** We compare UpFusion[†] (3D) to two single-view baselines and one sparse-view baseline (FORGE) on the GSO dataset. For each instance, single-view methods use only the image with the black border as input, whereas sparse-view methods use all input images. We can observe that UpFusion[†] (3D) while using 6 inputs views is able to better understand the 3D structure of the object than the single-view baselines. Moreover, it is able to incorporate information from the 6 inputs views much better than the sparse-view baseline.

a strong scaling similar to pose-aware methods that can geometrically identify relevant aspects of input images. Finally, while our work provided a possible path for 3D inference from unposed views by sidestepping the task of pose estimation, it remains an open question whether explicit pose inference for 3D estimation might be helpful in the long term.

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

## A   ADDITIONAL RESULTS

We visualize additional samples results from UpFusion (3D) for seen and unseen categories in Figures (7,8, 9) and Figure 10 respectively.

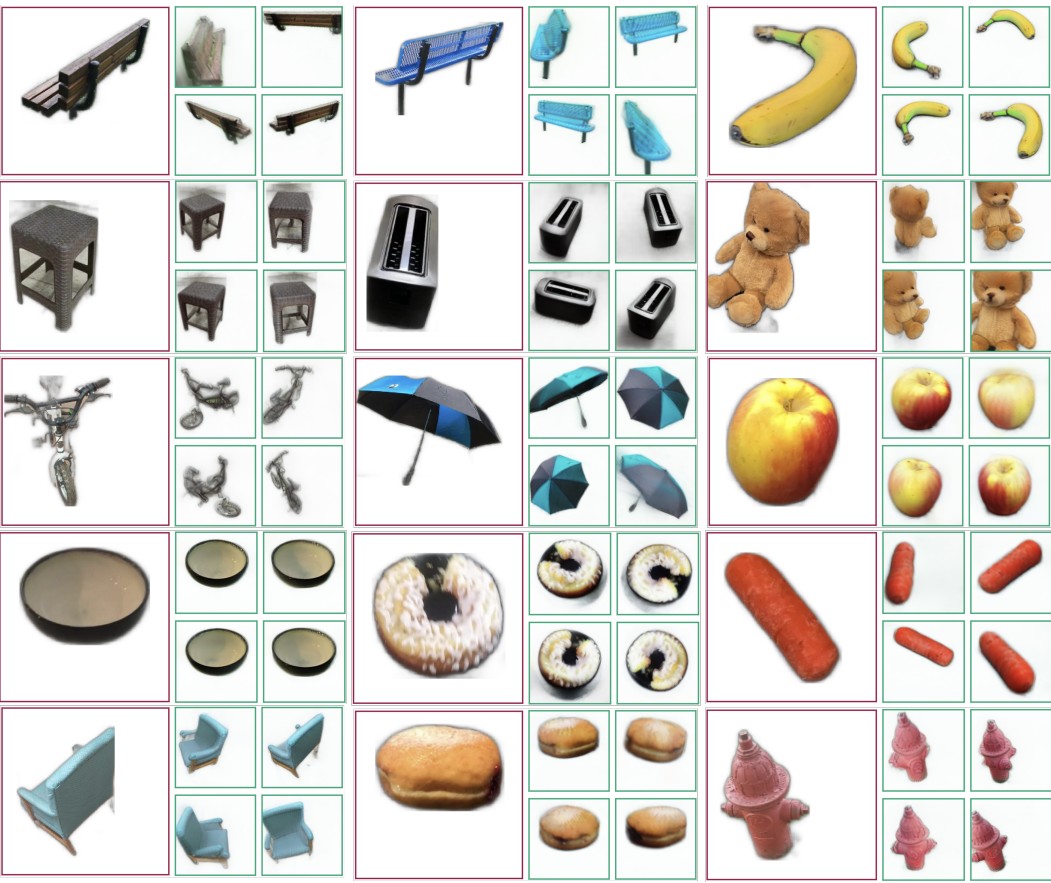

Figure 7: **Additional results with 1 input view.** We show results for UpFusion (3D) across different object categories given 1 input view (left), and show 4 novel view renderings (right).

## B   IMPLEMENTATION AND TRAINING DETAILS

### B.1   UPSRT

We use 8 encoder transformer blocks and 4 decoder transformer blocks in our architecture. For the feature extractor, we use features from DINOv2 (specifically, the *dinov2_vitb14*) model. We leverage the *key* facet from the attention block number 8 to extract features. We use sinusoidal positional encoding instead of learnable positional encoding for the camera and patch encoding. We also provide information about image intrinsics in the form of additional positional encoding. We trained our model for about 1M optimizer steps on 2 GPUs with a global batch size of 12.

### B.2   UPFUSION 2D

Following suggestions from the ControlNet Zhang & Agrawala (2023) repository, we start training our model with the Stable Diffusion decoder locked for a few iterations. Then, we resume training with the decoder unlocked. We train our model for about 1M optimizer steps on 2 GPUs with a global batch size of 8.

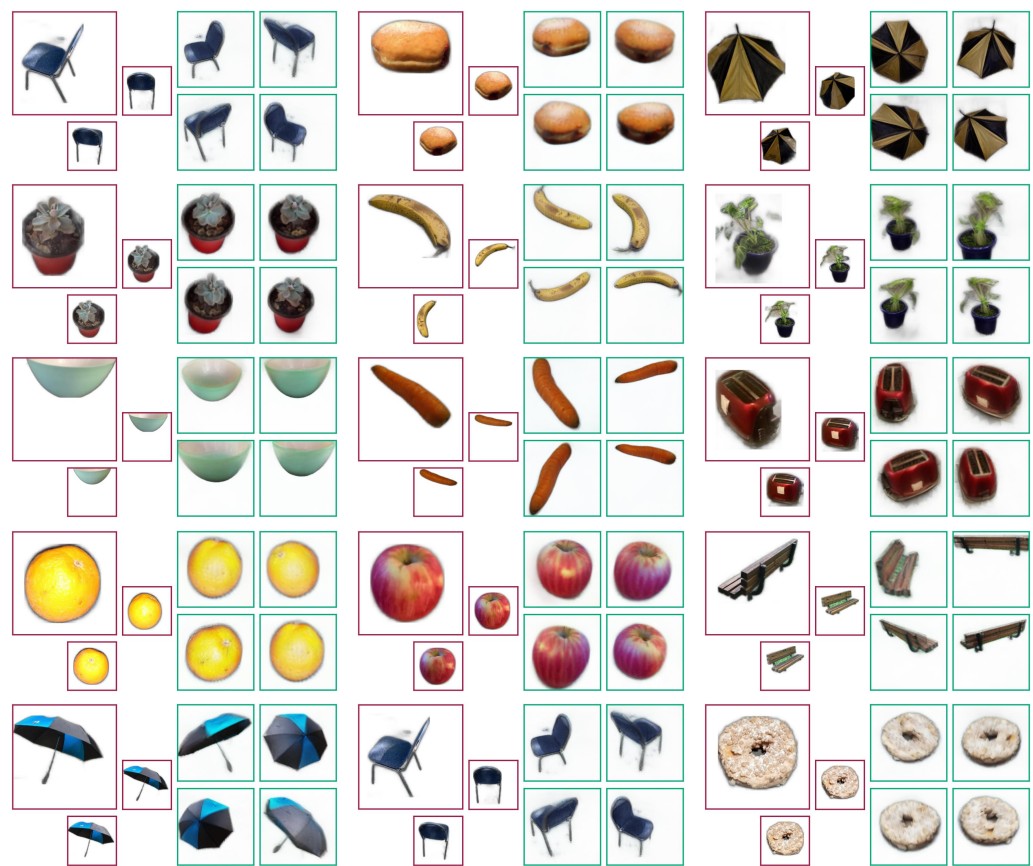

Figure 8: **Additional results with 3 input views.** We show results for UpFusion (3D) across different object categories given 3 input views (left), and show 4 novel view renderings (right).

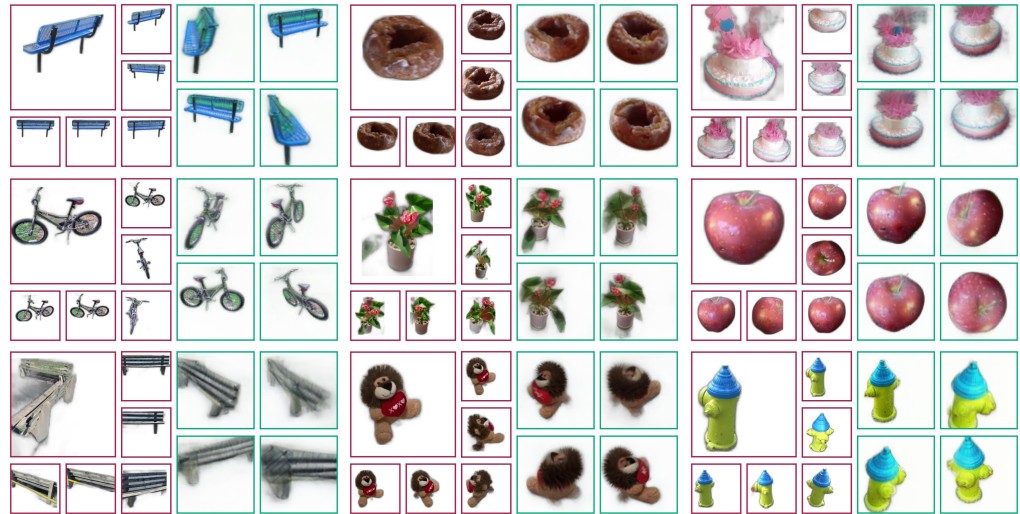

Figure 9: **Additional results with 6 input view.** We show results for UpFusion (3D) across different object categories given 6 input views (left), and show 4 novel view renderings (right).

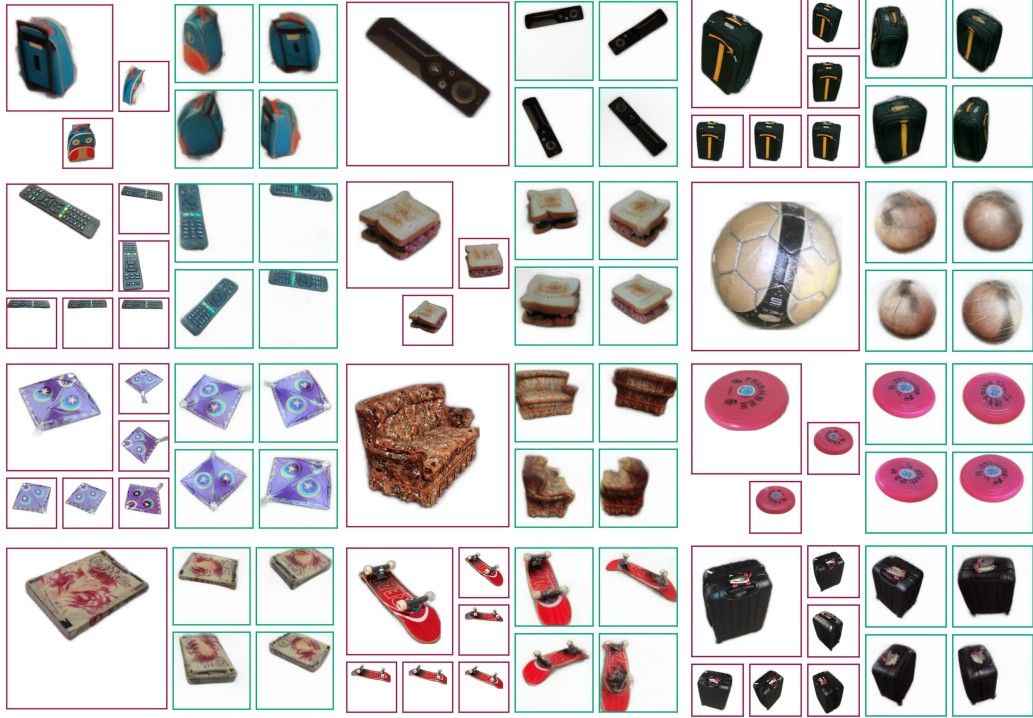

Figure 10: **Additional results for generalization beyond training categories.** We show results for UpFusion (3D) across object categories *not* seen in training. For each instance, we present the 1,3, or 6 unposed input views (left), as well as 4 novel view renderings (right). We observe that despite not being trained on these categories, UpFusion is able to accurately infer the underlying 3D structure and generate detailed novel views.

### B.3 UPFUSION 3D

For the multi-step diffusion model sampling, we use the DDIM (Song et al., 2020) sampler. Inspired by (Wang et al., 2023b), we use an annealed time schedule for optimizing our NeRF. For the first 300 iterations, we sample time steps corresponding to very high noise to enable the NeRF to quickly learn coarse level details. Overall, the NeRF is trained for 3000 iterations, which takes a little more than an hour on an A5000 GPU. We use the same regularization losses as of (Zhou & Tulsiani, 2023).

## C ADDITIONAL EXPERIMENTS

### C.1 ABLATING DIFFUSION CONDITIONING.

We empirically study the complementary benefits of the two forms of conditioning used. We illustrate the qualitative results in Figure 11 and quantitative results in Table 4. We find that both, the decoder features and the set-latent representations are complementary and both instrumental to UpFusion.

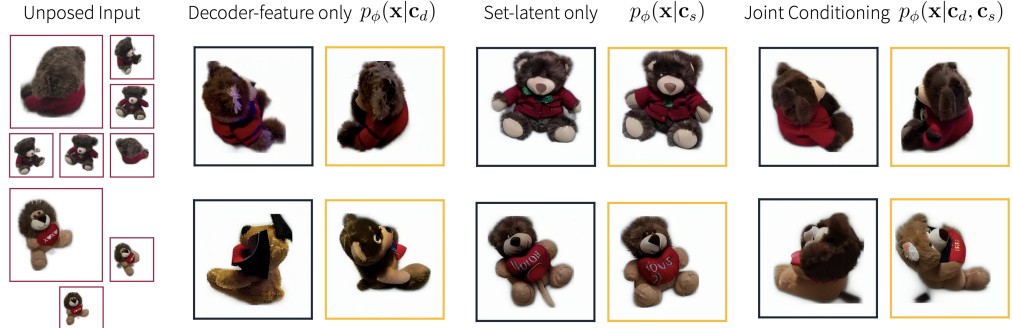

Figure 11: **Ablation of generative model conditioning.** Visualizations from category-specific models trained teddybears using varying conditioning for novel-view generation. We find that the model using only set-latent conditioning is unable to understand the query pose, while the one relying on only decoder features alters the object identity.

| Conditioning | PSNR-A ($\uparrow$) | | | SSIM-A ($\uparrow$) | | | LPIPS-A ($\downarrow$) | | |
|---|---|---|---|---|---|---|---|---|---|
| | 1V | 3V | 6V | 1V | 3V | 6V | 1V | 3V | 6V |
| DF Only | 14.65 | 15.28 | 15.53 | 0.63 | 0.64 | 0.64 | 0.32 | 0.30 | 0.30 |
| SLT Only | 13.15 | 13.38 | 13.49 | 0.60 | 0.60 | 0.60 | 0.36 | 0.35 | 0.35 |
| DF+SLT | **15.58** | **16.11** | **16.26** | **0.65** | **0.66** | **0.66** | **0.30** | **0.28** | **0.28** |

Table 4: **Ablation of generative model conditioning.** We ablate our conditional diffusion model with different conditional contexts. **DF** stands for decoder features $c_d$, and **SLT** stands for set-latent representations $c_s$. We train a category-specific generation model for this ablation on the teddybear category and report performance averaged across all test instances.

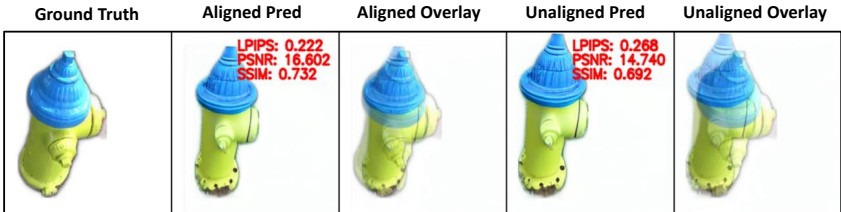

Figure 12: **Comparison of aligned and unaligned metric.** Conventional image reconstruction metrics are not well-suited to evaluate unposed view synthesis methods due to the inherent ambiguities between coordinate systems. We adopt aligned versions of these metrics by first performing optimized image warping. We illustrate the images and metrics with and without the alignment.

