# OpenReview forum: "UpFusion: Novel View Diffusion from Unposed Sparse View Observations"
_ICLR.cc/2024/Conference — Submitted to ICLR 2024_

### Official Review · Reviewer_rcbB · 2023-10-31

**Soundness:** 2 fair
**Presentation:** 3 good
**Contribution:** 2 fair
**Rating:** 5
**Confidence:** 5

**Summary:**

The authors introduce UpFusion, a view synthesis method derived from a collection of unposed images. The core design philosophy of UpFusion centers around the use of a Scene Representation Transformer, combined with a Diffusion Model to infuse intricate object details. Subsequently, instance-specific neural representations are introduced to achieve 3D-consistent rendering outcomes.

**Strengths:**

Scene Representation Transformer (SRT) is renowned as an effective neural renderer that can seamlessly generalize to the rendering of novel scenes and views. However, the SRT represents an image in the latent space, often resulting in blurry rendered outcomes, as evidenced in Fig 2 of this paper. UpFusion examines the constraints in SRT and suggests employing a following diffusion model and an instance-specific 3D representation to enrich the details. Specifically, the denoising diffusion model and a control net branch are employed to master a generative model for novel views of an object, and the instance-specific representation adheres to the paradigm in Score Distillation Sampling to extract a consistent 3D representation.

**Weaknesses:**

- The reviewer recommends that the authors emphasize the object-level NVS configuration in the title since the methodology chiefly addresses "objects" and the experiments were executed on the "CO3D" dataset.
- Object-level 3D generation (sourced from unposed images or a single image) remains a hot research topic. There exists a plethora of related papers [1,2,3, 4]. However, pivotal experimental comparisons with [1,2,3,5] are missing. Notably, single-view based NVS can ignore the requirment for camera poses: [1,2,3] all necessitate an object-specific representation, while [5] solely requires a forward-pass for view generation.
- What are the specifics regarding the training time for each instance? Considering [4] also employs a 3D representation to tackle a similar scenario, but does not incorporate the SRT and diffusion model, it's useful for the authors to showcase the merits of solely leveraging a 3D representation. Further, comparisons excluding the SRT/Diffusion model or contrasting it against [4] would be insightful.
- The CO3D dataset is characterized by various backgrounds, yet the authors omit the background modeling in the manuscript (possibly using the masks in CO3D). Even though object-level NVS publications typically sidestep background modeling, it's essential to accentuate this specific operations in the experimental framework.
- In terms of the claim 3D consistent generation, it would be useful if the authors could provide diverse rendered video of different objects, as well as producing metrics for your claim.

[1] One-2-3-45: Any Single Image to 3D Mesh in 45 Seconds without Per-Shape Optimization
[2] Make-It-3D: High-Fidelity 3D Creation from A Single Image with Diffusion Prior
[3] NeRDi: Single-View NeRF Synthesis with Language-Guided Diffusion as General Image Priors
[4] Few-View Object Reconstruction with Unknown Categories and Camera Poses
[5] Zero-1-to-3: Zero-shot One Image to 3D Object

**Questions:**

See the recommendated experiments in **Weaknesses** .

---

> ### Author Response · Authors · 2023-11-22
>
> We thank the reviewer for their valuable feedback. In this section, we elaborate on concerns that are not covered in the common rebuttal.
>
> > There exists a plethora of related papers [1,2,3, 4]. However, pivotal experimental comparisons with [1,2,3,5] are missing.
>
> We have updated the related works section to include discussion about more single-view methods, and will incorporate the reviewer feedback to emphasize the ‘object-centric’ of our presentation. Moreover, we have included comparisons against some state-of-the-art single-view baselines, details of which are elaborated in the common rebuttal section.
>
> > What are the specifics regarding the training time for each instance?
>
> The neural mode distillation stage currently takes a little more than an hour to optimize per instance. We have included these details in the supplementary material.
>
> > The authors omit the background modeling in the manuscript (possibly using the masks in CO3D)
>
> We do indeed use the masks provided in CO3D and sidestep modeling the background. We have clarified this point in the main text.
>
> > it would be useful if the authors could provide diverse rendered video of different objects, as well as producing metrics for your claim (of 3D consistency)
>
> We have included GIFs that showcase objects estimated by UpFusion 3D in this anonymized [website](https://iclr24-paper-4145.pages.dev/). Moreover, UpFusion 3D is 3D consistent by construction since we optimize a NeRF to represent an object (and NeRFs are 3D consistent).
>
> > Further, comparisons excluding the SRT/Diffusion model or contrasting it against [4] would be insightful
>
> As recommended by the reviewer, we include quantitative and qualitative comparisons against FORGE (see table 3 and figure 6; and also the common rebuttal). We note that our model significantly outperforms FORGE when using 6 input views, and we believe this is because our approach allows bypassing explicit pose prediction which can lead to inaccurate predictions.

---

> > ### Comment · Reviewer_rcbB · 2023-12-02
> > **Feedbacks from reviewer**
> >
> > Thank you for the authors' responses, particularly for including new experiments and visualizations. However, most of my concerns remain unresolved:
> >
> > - The rationale behind incorporating the keyword "Object" in the title.
> > - The omission of three-view comparisons in Table 3, as a three-view is a commonly adopted setting in sparse-view NVS.
> > - The uploaded GIFs exhibit only certain frames at a low FPS, and there are some artifacts at the boundaries. Clarifications on the reasons for these results should be provided.
> >
> > I would keep the original score.

---

### Official Review · Reviewer_UGEs · 2023-11-01

**Soundness:** 2 fair
**Presentation:** 2 fair
**Contribution:** 2 fair
**Rating:** 6
**Confidence:** 4

**Summary:**

The paper focuses on view synthesis from unposed images. Scene Representation Transformer, diffusion model, and controlnet branch are utilized to effectively perform the task on object-level novel view synthesis.

**Strengths:**

SRT + Diffusion are adopted to study the challenging task of novel view synthesis from unposed images.

**Weaknesses:**

The experiments are conducted on object-level generation. As the authors also mentioned in the related works section, single view image-to-3d is highly related to the task UpFusion trying to solve. Single view input can also be considered as input image without pose. As a result, I believe the contributions of UpFusion can be better justified when comparing to existing single view novel view synthesis works, for example [1].
Besides, a couple of references are missing [2] (also on Co3D dataset), [3][4] (SDS-based).

[1] Zero-1-to-3: Zero-shot one image to 3d object

[2] NeRDi: Single-View NeRF Synthesis with Language-Guided Diffusion as General Image Priors

[3] RealFusion: 360° Reconstruction of Any Object from a Single Image

[4] NeuralLift-360: Lifting An In-the-wild 2D Photo to A 3D Object with 360 views

**Questions:**

Please refer to the weaknesses.

---

> ### Author Response · Authors · 2023-11-22
>
> We thank the reviewer for their valuable feedback.  We have added the references pointed out to our discussion.
>
> > I believe the contributions of UpFusion can be better justified when comparing to existing single view novel view synthesis works
>
> We have included comparisons against some state-of-the-art single-view baselines, details of which are elaborated in the common rebuttal section. In particular, we find that our approach is competitive with the SoTA single-view methods given 1 input image, but can scale with more views to outperform the single-view baselines given additional (unposed) images.

---

### Official Review · Reviewer_yMjS · 2023-11-01

**Soundness:** 3 good
**Presentation:** 2 fair
**Contribution:** 2 fair
**Rating:** 5
**Confidence:** 4

**Summary:**

This paper proposes a framework for 3D-aware novel view synthesis given sparse 2D views without camera poses. Leveraging the features of the 2D views from an UpSRT encoder-decoder, it predicts a view-aligned spatial feature for the target view and a set-invariant feature, and feeds them to Stable Diffusion as conditioning inputs to generate the novel views. Moreover, it also incorporates an underlying 3D representation with NeRF to further enforce view consistency.

**Strengths:**

- The paper combines view-aligned spatial features and set-invariant features from unposed 2D views and leverages in novel-view diffusions, which sounds natural.
- Both qualitative and quantitative results show that the proposed framework outperforms the prior works not integrating these modules.

**Weaknesses:**

- Experimental comparisons with existing works seem limited. The following works are also related and could be discussed in the paper: [1,2,3,4,5]. Several of these works, as well as the single-/few-view NeRF synthesis works mentioned in the related work section, can be compared with the proposed method experimentally. Specifically, the single-view works also don't rely on relative poses, though the setup is not exactly the same as this paper, they can still be compared.
- Quantitatively, the UpFusion 3D model has much better numbers than the 2D model, but visually it loses a lot of geometric details compared to the 2D results. Is it limited by the representation power of the 3D NeRF? Or is it because the learned features are not very view-consistent?

[1] Ye, Yufei, Shubham Tulsiani, and Abhinav Gupta. "Shelf-supervised mesh prediction in the wild." Proceedings of the IEEE/CVF Conference on Computer Vision and Pattern Recognition. 2021.
[2] Deng, Congyue, et al. "Nerdi: Single-view nerf synthesis with language-guided diffusion as general image priors." Proceedings of the IEEE/CVF Conference on Computer Vision and Pattern Recognition. 2023.
[3] Tang, Junshu, et al. "Make-it-3d: High-fidelity 3d creation from a single image with diffusion prior." arXiv preprint arXiv:2303.14184 (2023).
[4] Liu, Minghua, et al. "One-2-3-45: Any single image to 3d mesh in 45 seconds without per-shape optimization." arXiv preprint arXiv:2306.16928 (2023).

**Questions:**

- Comparing the 2D and the 3D UpFusion model, I understand that the PSNR and SSIM of the 3D model are better, but why is the LPIPS also better? -- LPIPS is not a pixel-aligned metric, while visually the 2D results look cleaner and have much more details than the 3D ones.
- It seems that the generated views sometimes have inconsistent colors as the input view (e.g. the blue bench and the blue umbrella in Fig. 8, Appendix A). Is there any explanation for this?
- I wonder how the proposed method compares to this baseline: first running COLMAP to estimate the relative poses of the input views, and then running a pose-dependent 3D synthesis method?

---

> ### Author Response · Authors · 2023-11-22
>
> We thank the reviewer for their valuable feedback. In this section, we elaborate on concerns that are not covered in the common rebuttal.
>
>
> > The following works could be discussed in the paper: [1,2,3,4,5] .... and can be compared with the proposed method experimentally.
>
> We have updated the related works section to include discussion about more single-view methods. Moreover, we have included comparisons against some state-of-the-art single-view baselines, details of which are elaborated in the common rebuttal section. We hope this experiment illustrates the benefit of our approach which is competitive with the single-view methods given one view, but can benefit from additional (unposed) images unlike the single-view methods and outperform them in this scenario.
>
> > Quantitatively, the UpFusion 3D model has much better numbers than the 2D model, but visually it loses a lot of geometric details compared to the 2D results. ‘
>
>
> While we agree that individual images from Upfusion2D are more detailed, these are often spurious details which might actually detract from the performance. In contrast, UpFusion3D recovered a consistent 3D mode whose renderings are ‘likely’ from all viewpoints, and thus maybe more likely to match the input.
>
> > but why is the LPIPS also better? -- LPIPS is not a pixel-aligned metric, while visually the 2D results look cleaner and have much more details than the 3D ones
>
> LPIPS is technically pixel-aligned on a feature-grid, and does still care about alignment (although perhaps less than other metrics). We conjecture that even LPIPS for Upfusion3D is better because predicting spurious details is penalized more than predicting less detail. We would also point out that a similar trend is also observed in prior work (e.g. in SparseFusion, their full approach performs better than the underlying 2D diffusion model even in LPIPS).
>
> > It seems that the generated views sometimes have inconsistent colors as the input view (e.g. the blue bench and the blue umbrella in Fig. 8, Appendix A)
>
> This is an interesting observation! As of now we believe that these colors could perhaps be close in latent space and hence the neural mode distillation process gets stuck on a mode that represents a faithful rendering of an object albeit with colors that are slightly off. We also see similar artifacts (perhaps more pronounced) when distilling 3D representations from Zero-123 with the SJC.
>
>  > I wonder how the proposed method compares to this baseline: first running COLMAP to estimate the relative poses of the input views, and then running a pose-dependent 3D synthesis method?
>
> In table 1 of the paper, we had already included a baseline which uses RelPose++ to estimate poses (which is shown to be superior to COLMAP in the sparse-view setting) and then use an independent pose-dependent 3D synthesis method, SparseFusion. Both, this experiment and the added comparisons to FORGE highlight that accurately estimating this pose explicitly is a hard task, and that methods relying on this are not robust.

---

### Official Review · Reviewer_Egt9 · 2023-11-02

**Soundness:** 2 fair
**Presentation:** 3 good
**Contribution:** 2 fair
**Rating:** 5
**Confidence:** 5

**Summary:**

This paper presents UpFusion, a system that can generate novel views from a sparse set of uncalibrated multi-view images. Technically, UpFusion consists of two parts: 1) the first part is a modified UpSRT which encodes unposed images into a set representation and renders the feature maps for the target view, 2) while the second part is a diffusion-based ControlNet, which generates the novel view conditioned on the set representation and the decoded feature map.
During training stage, UpSRT and ControlNet are optimized separately.

**Strengths:**

+ The paper is well-written and well-structured. The problem setting is both innovative and ambitious, as it seeks to address two issues of considerable interest within the research community: 1) reconstruction from sparse views and 2) generation from unposed images, simultaneously.

+ The proposed method is intuitive, and this paper presents specific details and dedicated designs that are well-suited for the challenges inherent to the problem under investigation.

+ The empirical results demonstrate the method's remarkable performance in generating novel views from a few-shot unposed images when compared to the baseline approaches.

**Weaknesses:**

- The problem formulation lacks clarity. Without specifying poses from images, it becomes ambiguous to define the pose of a target view unless canonical poses are provided. However, canonicalization necessitates per-category calibration, and it has been observed that such methods are specific to certain categories. To further validate the effectiveness of the approach, it is recommended to test the pre-trained UpFusion on additional data domains, such as Blender, LLFF, or Shiny datasets [1].

- Empirical comparisons are insufficient in relevant baseline models. For the task of novel view synthesis, it is advisable to include comparisons with end-to-end pose optimization baselines, such as BARF [2] and NoPe-NeRF [3]. Since this paper asserts novel view generation from sparse views using diffusion, it would also be equitable to compare with state-of-the-art single-image-to-3D baselines such as Zero-123 [4].

- From a technical perspective, despite notable engineering efforts, the proposed method appears to be a combination of existing methods: SRT, ControlNet, and DreamFusion. It also structurally resembles GeNVS [5].

- The paper lacks a discussion and comparison with some relevant prior work, particularly with references [6] and [7].

[1] Wizadwongsa et al.NeX: Real-time View Synthesis with Neural Basis Expansion

[2] Lin et al., BARF: Bundle-Adjusting Neural Radiance Fields

[3] Bian et al., NoPe-NeRF: Optimising Neural Radiance Field with No Pose Prior

[4] Liu et al., Zero-1-to-3: Zero-shot One Image to 3D Object

[5] Chan et al., GeNVS: Generative Novel View Synthesis with 3D-Aware Diffusion Models

[6] Smith et al., FlowCam: Training Generalizable 3D Radiance Fields without Camera Poses via Pixel-Aligned Scene Flow

[7] Fu et al., MonoNeRF: Learning Generalizable NeRFs from Monocular Videos without Camera Poses

**Questions:**

1. The evaluation scheme proposed in Sec. 4.1.2 is designed to mitigate pose ambiguity. However, the enforcement of per-view alignment may introduce more confusion in the evaluation results, making it challenging to assess whether the proposed method can accurately generate views at specified camera poses and maintain smooth views along a camera trajectory. To enhance the evaluation methodology, it is recommended that the authors consider implementing a global alignment across all views collectively, rather than performing alignment on a per-frame basis.

2. Could the authors provide insights into the motivation behind making the specific modifications to UpSRT?

---

> ### Author Response · Authors · 2023-11-22
>
> We thank the reviewer for their timely feedback. In this section, we elaborate on concerns that are not covered in the common rebuttal.
>
> > Since this paper asserts novel view generation from sparse views using diffusion, it would also be equitable to compare with state-of-the-art single-image-to-3D baselines such as Zero-123.
>
> We have now included comparisons with state-of-the-art single-image-to-3D baselines. We have included details in the common rebuttal. We hope this experiment illustrates the benefit of our approach which is competitive with the single-view methods given one view, but can benefit from additional (unposed) images unlike the single-view methods and outperform them in this scenario. We hope these results on GSO also address the concern regarding evaluation on additional domains.
>
> > Without specifying poses from images, it becomes ambiguous to define the pose of a target view unless canonical poses are provided
>
> Following prior conventions e.g. UpSRT, we use the first input image as an anchor to define the orientation of the coordinate system. The pose of a target view is then defined w.r.t. this coordinate frame, thus making the task well-posed. While there is still a scale ambiguity in the reconstruction, we normalize our coordinate system during training to minimize this, and also allow the alignment during evaluation to not penalize methods for such ambiguity.
>
> We mention this choice of the coordinate frame in Sec 3.2, and would be happy to expand on this further in the text.
>
>  > For the task of novel view synthesis, it is advisable to include comparisons with end-to-end pose optimization baselines, such as BARF [2] and NoPe-NeRF [3] … The paper lacks a discussion and comparison with some relevant prior work, particularly with references [6] and [7]
>
> Thank you for these references. We have edited the related work to discuss these further. While these methods can estimate pose, they are not applicable in our setting as they operate on densely-sampled images e.g. NoPe-NeRF presents results on images from dense *trajectories*, and BARF requires ~100 images for a 360-degree inference. As perhaps a more directly applicable baseline, we have now added a comparison to FORGE (please see common response above) which explicitly infers sparse-view poses followed by reconstruction, and we find that our system is more robust.
>
> > To enhance the evaluation methodology, it is recommended that the authors consider implementing a global alignment across all views collectively, rather than performing alignment on a per-frame basis.
>
> We fully agree with the reviewer’s comment. In fact, we initially attempted gradient based optimization techniques to calculate a global alignment. However, in practice we found that it was not possible to obtain gradients to perform this computation across all baselines (e.g. differentiably transforming SparseFusion’s InstantNGP reconstructions was not feasible) and hence we settled for a per-frame alignment method which readily works across all baselines.
>
> However, in the additional results presented on GSO, we forego the aligned metrics and report the ‘regular’ metrics as the training data on Objaverse has origin-centered and normalized meshes, thus removing the ambiguity (unlike CO3D, where each object is in arbitrary coordinate frame).
>
> > Could the authors provide insights into the motivation behind making the specific modifications to UpSRT
>
> We use a pre-trained DINOv2 backbone (instead of a CNN) because DINO features have been shown to be informative for correspondences (which we assumed might help our downstream application).

---

### Author Response · Authors · 2023-11-22

We thank all the reviewers for their insightful suggestions and feedback. In this section, we address comments that are relevant to all reviewers. We address reviewer-specific concerns as direct comments to their respective reviews.

Following a common suggestion, we have added gifs showing 360-degree visualizations of our results (as well as baseline reconstruction) at this anonymous [website](https://iclr24-paper-4145.pages.dev/).

Another common suggestion by the reviewers was to add comparisons against single-view 3D approaches. To this end, we have now included comparisons of our method against two representative state-of-the-art single-view baselines, namely Zero-1-to-3 and One-2-3-45. We compare with two versions of Zero-1-to-3, the original version that (closer to ours) uses SJC for 3D distillation, and the highly optimized threestudio version (which uses additional tricks to aid 3D distillation). As these single-view models have been trained on a large synthetic dataset (Objaverse), we fine-tune our model on Objaverse renderings as well to allow for a fair comparison.

Additionally, based on comments from Reviewer rcbB, we also include comparisons against FORGE, a method that optimizes poses jointly with the objective of novel-view synthesis as an additional sparse-view baseline.

As all these methods evaluate their generalization capability on Google Scanned Objects, we perform our comparisons on 20 randomly sampled instances from GSO (with 32 evaluated views per instance).

The results of these are included in Table 3 in the main text, and are also summarized below. Qualitative comparisons are also added to the main text, as well as the [website](https://iclr24-paper-4145.pages.dev/).

| # Input Views   | Method   	 |  PSNR | SSIM | LPIPS |
|---------------|---------------|:-----:|:----:|:-----:|
| 1V | Zero-1-to-3 (SJC) | 18.72 | _0.90_ |  0.12 |
| 1V | Zero-1-to-3 (TS) 	 | _21.71_ | **0.91** |  _0.09_ |
| 1V | One-2-3-45		 | 17.77 | 0.87 |  0.15 |
| 1V | UpFusion$^†$ (3D) | 20.52 | 0.89 |  0.12 |
| 6V | FORGE     | 17.40 | 0.88 |  0.15 |
| 6V | UpFusion$^†$ (3D) | **22.51** | **0.91** | **0.08** |

We note that our model with 6 input views outperforms all baselines. More importantly, we can observe a clear improvement in performance when more views are provided to our method, which poses an advantage when compared to single-view baselines which cannot incorporate additional information.

We once again thank all the reviewers for their valuable suggestions. Additional reviewer specific comments are addressed below as individual replies to each review.

---

### Meta-Review · Area_Chair_QGi7 · 2023-12-11

**Metareview:**

The paper proposed a system, UpFusion, to synthesize novel views and 3D reconstruction from a sparse set of images without pose information.

The reviews are overall leaning negative (with three ratings marginally below and one marginally above the acceptance threshold).
Among the four reviews, only one reviewer lean to accept. Furthermore, the supporting review is considerably shorter than others.

The primary concerns are:
- unclear problem formulation,
- insufficient baseline comparisons (e.g., BARF, NoPe-NeRF, zero123)
- the technical method is a combination of SRT, ControlNet, and DreamFusion.

The authors responded to the reviewers' comments and provided additional quantitative results. Unfortunately, the reviewers did not respond and engage in the discussions. However, the AC reviewed the additional results and found that there is no clear advantages over baselines. For example, in 1V setting, the zero123 baseline actually performs better in all metrics. In 6 views setting, the proposed method perform similar/marginally better. The provided visual results also consist of many visual artifacts. The AC understands that single image 3D methods cannot leverage these additional unpose views.

Considering the marginal improvement, limited technical novelty (combination of existing approaches), and no clear supporting arguments among the reviewrers, the AC finds no ground to accept the paper.

**Justification For Why Not Higher Score:**

The evaluation is limited. The improvement is marginal.

**Justification For Why Not Lower Score:**

N/A

---

### Decision · Program_Chairs · 2024-01-16

Reject